# Molecular Diagnosis of Leishmaniasis: Quantification of Parasite Load by a Real-Time PCR Assay with High Sensitivity

**DOI:** 10.3390/pathogens10070865

**Published:** 2021-07-09

**Authors:** Germano Castelli, Federica Bruno, Stefano Reale, Simone Catanzaro, Viviana Valenza, Fabrizio Vitale

**Affiliations:** 1Centro di Referenza Nazionale per le Leishmaniosi (C.Re.Na.L.), OIE Leishmania Reference Laboratory, Istituto Zooprofilattico Sperimentale della Sicilia, Via Gino Marinuzzi 3, 90129 Palermo, Italy; germano.castelli@izssicilia.it (G.C.); simone.catanzaro@community.unipa.it (S.C.); viviana.valenza@you.unipa.it (V.V.); fabrizio.vitale@izssicilia.it (F.V.); 2Laboratorio di Tecnologie Diagnostiche Innovative (TDI), Istituto Zooprofilattico Sperimentale della Sicilia, Via Gino Marinuzzi 3, 90129 Palermo, Italy; stefano.reale@izssicilia.it

**Keywords:** *Leishmania infantum*, Real-Time PCR, molecular diagnosis, parasite load

## Abstract

Real-time PCR was developed to quantify *Leishmania infantum* kinetoplast DNA and optimized to achieve a sensitivity of 1 parasite/mL. For this purpose, we cloned the conserved kDNA fragment of 120 bp into competent cells and correlated them with serial dilutions of DNA extracted from reference parasite cultures calculating that a parasite cell contains approximately 36 molecules of kDNA. This assay was applied to estimate parasite load in clinical samples from visceral, cutaneous leishmaniasis patients and infected dogs and cats comparing with conventional diagnosis. The study aimed to propose a real-time PCR for the detection of *Leishmania* DNA from clinical samples trying to solve the diagnostic problems due to the low sensitivity of microscopic examination or the low predictive values of serology and resolve problems related to in vitro culture. The quantitative PCR assay in this study allowed detection of *Leishmania* DNA and quantification of considerably low parasite loads in samples that had been diagnosed negative by conventional techniques. In conclusion, this quantitative PCR can be used for the diagnosis of both human, canine and feline Leishmaniasis with high sensitivity and specificity, but also for evaluating treatment and the endpoint determination of leishmaniasis.

## 1. Introduction

*Leishmania* is a parasitic protozoa belonging to the family Trypanosomatidae, which can infect both humans and other mammals [1]. Leishmaniasis is an infectious disease that affects people, domestic and wild animals in temperate, subtropical, and tropical regions worldwide caused by diphasic protozoans of the genus *Leishmania* [2]. Dogs are considered the main reservoir of *Leishmania infantum* (*L. infantum*), developing canine Leishmaniosis (CanL) but there is clear evidence that some wild and synanthropic mammals and domestic cats can infect sand flies and they play a variable role in a reservoir system according to local and ecological peculiarities [3]. Cats also develop clinical disease, feline leishmaniasis (FeL) due to *L. infantum* infection and may have dermal as well as visceral manifestations of this infection [4,5]. They can therefore play a role as additional reservoir hosts of *L. infantum* and in a “One Health” perspective; preventative measures should be taken in this species based on epidemiological data [6]. In humans as in canine leishmaniosis, can produce a wide spectrum of lesions and clinical signs, but some of them are considered highly suggestive of the disease in endemic areas (lymphadenomegaly, skin and ocular lesions, epistaxis, weight loss, signs of renal failure) [2,5,7]. Human leishmaniasis is a complex disease, with heterogeneous clinical manifestations ranging from asymptomatic infections to lesions at cutaneous sites (cutaneous leishmaniasis, CL), mucosal sites (mucocutaneous leishmaniasis, MCL), or in visceral organs (visceral leishmaniasis, VL), depending on the species and host characteristics [8,9]. The diagnosis of leishmaniasis is confirmed by the demonstration of parasite in tissues of relevance by light microscopic examination of the stained specimen, in vitro culture, indirectly through positive serology and nucleic acid demonstration [9,10,11,12,13,14,15,16,17]. Clinical status and a positive immunological test can be prognostic, but the diagnostic ‘gold standard’ is parasite identification by isolation on a specific culture medium [18,19]. Moreover, microscopic examination of tissue biopsy specimens, bone marrow, or splenic aspirates, are important to help the diagnostic approach [10,18,19,20]. Each of these diagnostic techniques has advantages and shortcomings, and the diagnostic methods should be selected in light of these considerations. In particular, the use of molecular techniques has become increasingly relevant due to their high sensitivity, specificity, and possible application to a variety of clinical samples [21]. Among them, real-time PCR, also named quantitative PCR (qPCR), has become increasingly popular recently since it is fast, has a broad dynamic range, and cross-contamination is drastically reduced because there is no need to open reaction tubes for post-PCR analyses [22]. The qPCR relies on an analysis of fluorescent signals produced during amplification. Fluorescence can be generated by using intercalating fluorescent dyes (e.g., SYBR Green) or fluorescent probes (e.g., TaqMan^®^) [21,23,24]. The PCR sensitivity is highest shortly after infection in asymptomatic dogs when serology sensitivity is low [25]. The seroconversion shows a strong positive correlation with the infectiousness of dogs [26,27] and precedes the appearance of clinical signs [2]. Qualitative PCR, while being sensitive and specific, does not provide information about changes in the parasitic burden in biological samples, and results from this molecular technique are not adequate in clinical trials designed for comparing the efficacy of different treatment protocols [28,29]. Other assays were developed to estimate the parasite load through absolute or relative quantification.

Conversely, Real-Time PCR allows the quantification of the nucleic acid template by analyzing the kinetics of PCR during the cycles and can rapidly quantify DNA [30,31,32] based on a calibration curve [33,34]. *Leishmania spp*. have 35 to 36 chromosomes in their nuclear genomes [35,36], and it is characterized by both mini and maxi circles kinetoplast DNA structures [37]. Various chromosomal and kinetoplast DNA sequences have been evaluated as targets for diagnostic systems and the extra-chromosomal circles showed always the best sensitivity and represent currently the most employed target with worldwide diffusion [38,39]. The use of high-copy 18S and minicircle kDNA sequences as a target usually allows amplification at the genus or subgenus level, due to their conserved sequences or to the heterogeneity of minicircle classes [21]. A qPCR, targeted to a 120 bp long kDNA consensus sequence, was employed at the National Reference Centre for Leishmaniasis for *L. infantum* detection [40]. The quantitative test was based on a calibration curve and allows estimation of the *Leishmania* parasite load for mL in the analyzed samples [41]. Comparative quantification experiments have been performed for parasite counting in several samples.

The present study aims to evaluate the potential use of the conserved motif of kDNA minicircles in a Taqman-based real-time PCR approach, to discriminate between *L. infantum* from the other species and therefore identify correlation with the clinical status of the subjects, thus allowing us to discriminate between symptomatic patients, cured patients, and asymptomatic carriers. Furthermore, in the present paper, we research the equivalence between the copy number of kDNA target and serial dilutions of the cultivated *Leishmania* cells to quantify the kDNA circle present in each parasite cells. For this purpose, we clone the 120 bp kDNA conserved fragment into competent cells and correlate them with serial dilutions of *L. infantum* DNA extracted from reference parasite cultures using Real-Time PCR method. In the current study, we use the assay to assess *L. infantum* loads serum, blood and bone marrow samples from VL patients; tissue/biopsies from CL patients, and blood, lymph node needle aspiration, and serum are collected from dogs and cats. The samples collected are used for both conventional and molecular diagnoses.

## 2. Results

This section may be divided into subheadings. It should provide a concise and precise description of the experimental results, their interpretation, as well as the experimental conclusions that can be drawn.

### 2.1. Sensitivity and Specificity of the Real-Time PCR Assay

The sensitivity of the qPCR reaction was tested by using serial dilutions of *L. infantum* parasite DNA extracted from a known number of parasites. Detection of the kinetoplast DNA of *L. infantum* reached the level of 1 parasite per mL in the reaction tube with a dynamic range of 10^6^. This sensitivity allowed a detection limit of 1 parasite DNA/mL of reaction, taking into account the amount of biological sample (2 µL of sample DNA) and the elution volume of the extracted DNA (100 µL). We tested the reproducibility of qPCR from 10^6^ parasites/mL to 0.1 parasite/mL by testing each sample DNA 10 times, the standard curves (Figure 1) were characterized by a correlation coefficient (r 2) of 0.99, and slopes of –3.13 (mean) indicating a high amplification efficiency (2.08) (2 would indicate 100% PCR efficiency). To verify the specificity of the assay, PCR experiments were performed testing a collection panel of DNAs belonging to different *Leishmania* species: *L. aethiopica* (MHOM/ES/72/L100), *L. braziliensis* (MHOM/BR/75/M2904), *L. major* (MHOM/SU/73/5ASKH), *L. tropica* (MHOM/SU/74/K27), and *L. amazonensis* (IFLA/Br/67/PH8). Assay specificity was also tested in the presence of DNA from pathogens other than *Leishmania* such as *Trypanosoma cruzi*, *Ehrlichia, Babesia*, and *Theileria* and no amplification was observed (data not shown).

### 2.2. Cloning and Correspondence Leishmania/mL-kDNA Copy Number

The recombinant plasmid was extracted at 2.9 × 10^10^ copy/µL and it was used for 6 decimal serial dilution points. Real-time PCR was employed both on the recombinant plasmid and on the DNA extracted from the cultivated strain to compare the signals of the respective serial dilutions. Parasites were quantified by the comparative Ct method, and at Ct = 20 value we obtained the cross point were the *Leishmania*/mL and kDNA copy number were equivalent. Because at this point 1.58 × 10^5^ *Leishmania*/mL corresponded to 5.8 × 10^6^ plasmid copy number, we calculated that one parasite cell contains about 36 kDNA minicircles molecules.

### 2.3. Analysis of Parasite Load in Clinical Samples

Table 1 shows the results of the indirect immunofluorescent assay (IFAT), Parasite culture (PC) and microscopic examination (MH) and qPCR conducted on clinical samples derived from VL (*n* = 15), CL (*n* = 15), CanL (*n* = 30) and FeL (*n* = 20).

Evaluation of parasite load was carried out in clinical samples of VL patients and CL patients at pre-treatment (n = 15) post-treatment stage (n = 15) respectively. At the time of diagnosis, the evaluation of peripheral blood/serum in VL showed a molecular positivity in 13/15 patients compared to conventional diagnostic methods: IFAT (13/15), PC (4/15), and MH (13/15). In VL patients, bone marrow was the matrix of choice for molecular diagnosis of *Leishmania* with a positivity percentage of 100% (15/15). With a conventional diagnosis, the positivity rate has also increased: 53.3% with PC (8/15) and 86.6% with MH (13/15) respectively.

The evaluation of parasite load in tissue/skin biopsy revealed a 100% real-time PCR positivity in CL patients in comparison with 53.3% (8/15) and 86.6% (13/15) of positivity obtained with PC and MH, respectively. During weekly follow-up of VL and CL patients, molecular investigations performed on bone marrow and tissue/skin biopsy showed a progressive reduction of parasite load (data not shown). Figure 2 showed the parasite load by real-time PCR in human clinical samples before starting treatment and one day after the last dose of treatment. In CanL, the evaluation of peripheral blood/serum showed positivity of 19/30 with qPCR, similar to IFAT methods, instead, PC and MH detected a lower positivity (16.6%). The positivity rate increased when using the popliteal lymph node as a biological matrix, in fact, we obtained 30/30 (100%) positivity with qPCR compared to 16/30 (53.3%) and 17/30 (56.6%) positivity obtained with PC and MH respectively. In feline patients, the positivity rate increased when using the popliteal lymph node with 20/20 (100%) positive in qPCR with respect to both 1/20 (5%) positive obtained with PC and MH. Using peripheral blood/serum the positivity rate was considerably lower with qPCR (1/20, 5%) than IFAT, which detected a higher number of positives (8/20, 40%).

Based on clinical evaluation, dogs were distributed to the asymptomatic group (presenting no clinical signs; n = 10) or the symptomatic group, presenting at least one symptom of CanL including weight loss, lymphadenopathy, dry exfoliative dermatitis, ulcers, periorbital alopecia, diffuse alopecia, ocular signs, or onychogryphosis; n = 20 (Table 2).

In the symptomatic group, 18/20 showed *Leishmania* antibody titers ≥160 and of these dogs, 15/20 showed MH and PC positivity. Parasite load obtained on all symptomatic dogs by qPCR in popliteal lymph node samples was higher, from 50 to 1.000.000 *Leishmania*/mL, than the parasite load in the blood. In particular, 2 symptomatic dogs (IDs: 7 and 8) were negative for both blood qPCR and all conventional methods, but they presented 50 and 100 *Leishmania*/mL by qPCR in the popliteal lymph node.

Whereas in the asymptomatic group, only one dog (ID = 23) showed IFAT positivity 1:160, 7/20 dogs had a ‘suspicious’ antibody titer (1:40 and 1:80) and 2/20 were serologically negative. Dog ID 23 showed the only positivity of the asymptomatic group for PC and 120 *Leishmania*/mL in the lymph node matrix and 10 *Leishmania*/mL in the blood sample by qPCR. Parasite load obtained with qPCR in popliteal lymph node samples showed a lower positivity than the symptomatic group, from 10 to 120 *Leishmania*/mL. Regarding the analysis carried out on the feline species, all 20 cats showed clinical signs of FeL: weight loss, lymphadenopathy, dry exfoliative dermatitis, ulcers, periorbital alopecia, diffuse alopecia, ocular signs, or onychogryphosis (Table 2). Serological investigation showed a positive antibody titer (≥1:160) in 8/20 cats, 6/20 had suspect titers (1:80) and 6/20 were negative. PC and MH methods showed negative results in all cats except for sample ID 36, which showed only in the lymph node needle aspiration matrix a positivity by conventional methods. The qPCR conducted on all needle aspiration matrices detected a parasite load with values from 5 to 980 *Leishmania*/mL while the molecular analysis on blood showed positivity of 2/20 cats.

## 3. Discussion

Several methods and tools have been developed over recent years for the detection, quantification, and identification of the parasite of the genus *Leishmania* [15,19,20,42,43]. Although advances in these methods have improved the sensitivity and specificity of leishmaniasis diagnosis, there are still some challenges to be overcome. For instance, developing affordable, fast, and accessible tests that can define *Leishmania* species will be a turning point in diagnosis, since the discrimination of species has significant importance for prognosis and species-specific treatments [20]. Parasitological and microscopic examinations are broadly used; although highly specific, they present insufficient sensitivity and do not provide *Leishmania* species identification [19,20]. Furthermore, biopsies can be very invasive and life-threatening, such as in some patients with VL [17,20,44]. In vitro cultivation is rarely used in routine clinical practice, given that they are generally only available in leishmaniasis diagnosis reference centers and the results can take weeks to be delivered [20]. Over the last decade, a variety of real-time PCR assays have been developed in the search for appropriate tools for the detection of different *Leishmania* species [17,19,43,45]. The efficacy of those assays has varied, however, depending on the copy number of the target sequence and the specimen type measuring the amount of DNA generated by monitoring the amplification of a specific target during each PCR cycle [46,47,48]. In this study, we have designed a quantitative real-time capable of detecting the parasite *L. infantum* using the target kinetoplast sequence. All Kinetoplastid flagellates possess a single mitochondrial genome known as the kinetoplast DNA (kDNA), which consists of several thousand circular DNA molecules linked together in a concatenated network [47,49]. A particular characteristic of the Kinetoplastida order is the mitochondrial DNA network organized in 20–50 maxicircles and 10,000–20,000 kDNA minicircles [38,47,48,50]. The conserved region of these minicircles has been used as a diagnostic PCR target since the 1990s [21,50,51]. Many authors have used several genomic targets, other than the kDNA minicircle conserved region for the diagnostic purpose [51], however, these assays may be less sensitive due to the lack of multiple copies of target sequence per cell. In this study, we used a qPCR based on the kDNA sequence regions conserved by Vitale et al. (2004) [40] with some modifications. This quantitative PCR assay allows highly sensitive and reproducible quantification of the parasite burden over a wide range (at least six logs of parasite concentrations). The assay was based on TaqMan chemistry to detect in real-time the molecules produced during the PCR and showed a specificity towards the *L. infantum* species, in fact, the analysis performed on different *Leishmania* species: *L. aethiopica* (MHOM/ES/72/L100), *L. braziliensis* (MHOM/BR/75/M2904), *L. major* (MHOM/SU/73/5ASKH), *L. tropica* (MHOM/SU/74/K27), *L. amazonensis* (IFLA/Br/67/PH8), other pathogens as *Trypanosoma cruzi, Ehrlichia, Babesia,* and *Theileria* showed no amplification. The application of PCR in the identification of *L. infantum* has great potential as a valuable tool in the detection of parasite DNA and the evidence suggests that PCR can identify the species of parasite involved in the old world [52]. However, future studies should be carried out to further explore this issue. There is no single molecular gold standard diagnostic test, for all regions of the world because there are changes in the DNA sequences within a species according to their origin. The qPCR assays for leishmaniasis that amplify species-specific DNA sequences are not appropriate for regions with sympatric *Leishmania* species. In fact, even in areas where *Leishmania* species are known, the presence of polymorphisms and variations in the copy number of target sequences (e.g., minicircle classes), could interfere with the performance of qPCR assays [21]. To rule out false negatives or co-infections with other *Leishmania* species, the species-specific qPCR should be combined with a genus-specific assay (broad range assay). Therefore, approaches based on initial amplification of genus-specific sequences followed by assays for species differentiation/typing should be useful [21].

The use of this qPCR obtained a LOD value of 1 parasite per reaction: the very high sensitivity is due to the high copy number of the target minicircle kDNA, which is present at about 10,000 copies per parasite. Furthermore, we assessed copy numbers of kDNA target is present in each *Leishmania* cell: the cloned kDNA fragment extract was compared with *Leishmania* parasite load, in a range spanning through 7-log serial dilutions. We obtained a linear increase of the Ct value both for plasmid kDNA and DNA equivalent to parasite/mL calculating that one parasite cell contains about 36 kDNA minicircles molecules. This fact could explain the sensitivity of the molecular approach based on the starting copy number of the marker for the real-time PCR. With plasmid quantification, we can gain valuable insight into the copy number of the cloned target although it seems more logical to use the kDNA primer set for quantification in terms of parasite/mL in biological matrices to provide diagnostic information on an individual patient for example to follow the clinic and response to therapy. Absolute quantification by Real-time PCR was capable of quantifying parasite numbers in clinical or experimental samples, by comparison to a standard curve obtained from parasite DNA. However, the use of kDNA for quantification is complicated by fact that *Leishmania* species contain multiple copies of both minicircle and maxicircle [21,37,47,48,53,54,55]. It is not known whether copy numbers differ between the *Leishmania* species or between parasite stages. However, the average load of the kinetoplast molecules in the infected subject can be used to measure the parasite load at different moments or for a therapy approach.

In this study, we evaluated the diagnostic accuracy of the real-time PCR across multiple presentations associated with *L. infantum* infection. The high sensitivity and specificity of this method suggest the possibility of using it to diagnose CL, VL, CanL, and FeL, endemic diseases in Italy [4,9,56,57]. *Leishmania infantum* DNA was detected (%) in different clinical samples identifying the parasite load.

Indeed, biological matrices, including blood, bone marrow aspirate, lymph node needle aspiration, biopsies/tissues of VL, CL, CanL, and FeL were analyzed with the qPCR assay.

The analysis in VL patients showed that bone marrow was the matrix of choice for qPCR with a 100% positivity compared to conventional methods (MH, PC) while blood showed positivity to qPCR comparable to IFAT. Importantly, qPCR represents a valid approach in endemic areas to detect parasite load especially in immunocompromised patients (e.g., HIV co-infection and transplanted patients). Molecular diagnosis performed on the tissues/biopsies of the 15 cl patients revealed that it is the approach with a higher sensitivity and specificity than conventional methods. Various studies of South and Central American CL caused by parasites of the *L. braziliensis* and *L. mexicana* complexes have compared the kDNA PCR diagnosis with conventional techniques [58]. With the exception of a few cases, PCR-based assays were determined to be significantly more sensitive than the classical parasitological methods of diagnosis [53,59,60,61]. These results are supported by some studies of Old World CL caused by *L. tropica* in Iran [62], *L. major* in Sudan [63], and *L. aethiopica* in Ethiopia [64]. Our study confirms these findings, showing that the kDNA qPCR of *L. infantum* is superior to parasitological methods for the diagnosis of CL, identifying additional patients missed by either microscopic examination or culture. In addition, the post-treatment results of VL and CL patients indicate the usefulness of the kDNA qPCR of *L. infantum* as a diagnostic tool to monitor therapeutic response. In this study, we also confirmed parasite positivity of canine and feline samples previously assayed by conventional diagnosis (MH, PC, and IFAT), found 63.3% of CanL and 40% FeL positivity. The qPCR detected *L. infantum* presence, especially in lymph node needle aspiration, with positivity rates of 100% in dogs and cats, demonstrating that this biological matrix is of choice. The lymph node-qPCR method showed a higher sensitivity to that obtained using blood samples, according to many authors, blood was the sample with the lowest sensitivity [54]. In Table 3 the authors showed a correlation between clinical symptomatology of CanL and all laboratory investigations, but dogs IDs 7 and 8, showing negative results for IFAT, MH, and PC, were positive only on lymph node qPCR detecting the presence of the parasite. Unfortunately, direct microscopic observation of stained smears or indirect culturing of samples were highly specific but not very sensitive [65,66] and Mancianti et al., [67] stated that IFAT could fail in the early stages of the disease when IgG are not yet detectable. Furthermore, asymptomatic patients affected by CanL showed an indirect absence of the parasite (antibody titer ≤160) but the presence of the target sequence by qPCR, with low parasites load. Serologic tests are the standard tools for identifying *Leishmania* antibody titer, but the difficulties associated with interpreting seroprevalence data are different: (i) serology is prone to non-specific cross-reactions [68]; (ii) there may be a delay between infection and seroconversion, and a fraction of infected patients may not seroconvert (e.g., because of innate resistance) [60]; (iii) seroconversion may not be permanent (e.g., because of the development of the humoral or cell-mediated immune response) [25]. Thus, the sensitivity and specificity of serologic tests can vary considerably and may underestimate the incidence of disease [25,69]. Microscopic examination (MH) has limited sensitivity, particularly in paucinfections. While in vitro cultures, despite being considered the diagnostic gold standard, are laborious, expensive, and susceptible to microbiologic contamination [10,17,18,21]. Of the 20 symptomatic cats, all showed a parasite load at qPCR that did not correlate with conventional methods, in fact, 6/20 showed IFAT titer 1:80, and 6/20 negative titers. Therefore, as previously shown it seems that there was no agreement between the results obtained by IFAT and qPCR [4,70]. This is a common finding in cats from *L. infantum* endemic areas, where discordant results between serologic and molecular techniques have been reported previously [4,71,72,73]. As hypothesized before, the negative/suspect results obtained in IFAT could reflect an ineffective immune system response or could be explained by the absence of antibody production during an early stage of infection. For this reason, both direct and indirect techniques should be performed in epidemiological studies [3,4,6]. More importantly, this qPCR test can be used not only to diagnose the current infection of patients with CL, VL, CanL, and FeL but also to assess the severity and efficacy of leishmaniasis treatment. Indeed, the parasite load detected by qPCR correlated with the severity of infection in terms of lesion size (CL) or the serologic immunoglobulin levels or splenomegaly tested by ultrasonography in visceral leishmaniasis or the history during the clinical examination [17]. Detection and identification of *Leishmania* species are essential for accuracy, rapid, and sensitive diagnosis of leishmaniasis, which has a great influence on effective treatment and control measures [17,19,20,74,75,76]. Therefore, knowledge of the different methodologies and their objectives is crucial.

We aimed to propose a qPCR for the detection of *Leishmania* DNA from clinical samples trying to solve the diagnostic problems due to the low sensitivity of microscopic examination or the low predictive values of serology, whose results can be affected by either persistent antibodies (false positive) or immunosuppression (false negative) and resolve problems related to in vitro culture [21,45,46]. The real-time PCR assay in this study allowed detection of *L. infantum* DNA and quantification of considerably low parasite loads in samples from VL, CL, CanL, and FeL patients who were diagnosed negative by conventional techniques. Thus, a method of diagnosis that is sensitive enough to detect low levels of the parasite in asymptomatic or early symptomatic infection could be applied on the large scale in endemic regions in all Mediterranean areas. In conclusion, the real-time PCR assay we described could be easily implemented to change the paradigm of routine laboratory diagnosis of leishmaniasis and its related complications.

## 4. Materials and Methods

### 4.1. Parasites

Promastigotes of *Leishmania (L.) infantum* (MHOM/TN/80/IPT1), *L. (V.) braziliensis* (MHOM/BR/1975/M2904), *L. (L.) amazonensis* (IFLA/BR/1967/PH8), *L. (L.) aethiopica* (MHOM/ES/1972/L100), *L. (L.) major* (MHOM/SU/1973/5ASKH), *L.* (*L.*) *tropica* (MHOM/SU/1974/K27) and epimastigotes from *Trypanosoma cruzi* (C DM28c strains), provided by the *Leishmania* collection from C.Re.Na.L. (IZSSI), Italy, were cultivated in RPMI-PY medium [77], which consisted of RPMI 1640 (Sigma R0883) supplemented with an equal volume of Pepton-yeast medium, 10% fetal bovine serum (FBS), 1% glutamine, 250 mg/mL gentamicin and 500 mg/mL of 5-fluorocytosine. The strains were routinely cultivated at 25 °C in RPMI-PY medium. After 3 to 5 days, promastigotes were harvested, washed, and resuspended in phosphate-buffered saline (PBS) pH 7.2.

### 4.2. Clinical Samples

The study comprised 30 dogs, 20 cats, 30 humans living in Sicily, which is endemic for *L. infantum* [56]. Fifteen human patients showing forms of CL were examined at the Section of Dermatology of the University of Palermo; on the other hand, 15 patients affected by VL were examined at the Infectious Disease Department of the University of Palermo. Patients with ongoing VL presented with symptoms such as persistent fever, splenomegaly, and/or hepatomegaly while cutaneous lesions were usually erythematous papules, slightly itchy, slow-growing, which subsequently progresses into nodules, then ulcers, over a few weeks/month. Patients of VL were given treatment with liposomal amphotericin B (AmBisome^®^) 3 mg/kg administered intravenous route (IV) once daily days 1–5, 14, and 21. CL cases were treated with pentavalent antimonial (SbV) therapy, the standard daily dose was 20 mg of SbV per kg, administered IV or intramuscular route (IM). The diagnosis of VL and CL was made by microscopic identification of the intracellular form (amastigote) in stained sections from bone marrow/skins biopsy, real-time PCR, and serological tests. For some patients, adjustment of the daily dose or the duration of therapy could be indicated. No standard treatment regimen had been established, various regimens had been used depending in part on the size, characteristics of the lesions, and immunological status. All patients were then followed for follow-up visits at weekly intervals, at which clinical manifestations were reassessed and biological samples were collected and evaluated for conventional and molecular diagnosis. Clinical samples were taken before starting treatment and one day after the last dose of treatment for qPCR evaluation.

In addition, the study included dogs and cats from veterinary medical centers. Serum, blood, and bone marrow samples were collected from VL patients; tissue/biopsies were collected from CL patients, blood, lymph node needle aspiration, and serum was collected from dogs and cats.

The samples collected were used for both conventional and molecular diagnoses. Clinical samples of VL and CL were taken before starting treatment and one day after the last dose of treatment.

### 4.3. Conventional Diagnosis

All samples tested came from patients whose *Leishmania* infection status was explored for different reasons such as exclusion or confirmation of the disease, early diagnosis of relapse in immunocompromised patients, survey after therapy. Diagnosis of the disease relies on relevant clinical symptoms, a high level of specific antibodies, and the presence of the parasite in bone marrow or blood, detected by conventional (microscopy and culture). A survey of treated patients or at-risk patients relies on the serological survey and a direct search for *Leishmania* performed on blood samples when one or more clinical symptoms occur. Parasitological examinations involved the analysis of Giemsa stained slides containing bone marrow smears or imprints of skin fragments for the detection of amastigote forms, and bone marrow culture in RPMI-PY [77]. The tubes were incubated at 24 °C, and an aliquot of the liquid phase of the medium was transferred to a fresh medium every week. Plates were visualized for promastigote growth with an inverted microscope at 200× magnification every week for 1 month. Serological examinations included IFAT assays carried specific antibodies to *L. infantum* were detected using the IFAT against in-house cultured promastigotes following Office International des Epizooties (OIE) Terrestrial Manual protocol [68]. *Leishmania* strain was used as an antigen fixed on multispot microscope slides (Bio-Merieux, Marcy L’Etoile, France) in an acetone bath. The canine sera were prepared by serial 2-fold dilutions (1:40 to 1:5120), while human and feline sera were prepared by serial 2-fold dilutions (1:80 to 1:5120) in phosphate-buffered saline (PBS), pH 7.2, and added to the antigen-coated wells. The slides were incubated for 30 min at 37 °C. Positive and negative controls were included in each series of analyzed samples. The slides were subjected to three consecutive washes in PBS (10 min each) and incubated with fluorescein-labeled goat anti-species immunoglobulin (working anti-cat, anti-human and anti-dog IgG (whole molecule)—FITC antibody produced in goat, Sigma Aldrich, Saint Louis, MO, USA) were added to wells and incubated for 30 min at 37 °C. The slides were washed three times (10 min each) in PBS and the sera reactivity was using a Leica DM 4000B fluorescence microscope (Leica, Heerbrugg, Switzerland) (40× magnification). The serum samples that showed reactivity were subsequently subjected to analysis at dilutions 1/40 and 1/5120. The samples that showed titers of ≥1:160 were considered positive [3,68].

### 4.4. DNA Extraction of Promastigote Cultures and Clinical Samples

The cultivated strains were harvested, washed twice with NaCl 0.3%, and centrifuged. The total DNA was extracted from the promastigotes according to as described in Castelli et al. (2020) [56] as follows: the pellet was lysed by heating at 96 °C for 20 min with 400 μL of a mixture containing 1% Tween 20 (Sigma, St. Louis, MO, USA), 1% Nonidet P-40 (Sigma, Tokyo, Japan) and 20% Chelex resin (Bio-Rad, Hercules, CA, USA) prepared in sterile distilled water. The mixture was centrifuged at 14,000× *g* for 10 min at 4 °C and the DNA-containing upper phase was then collected and stored at –20 °C until used. DNA was extracted from different clinical samples using PureLink™ Genomic DNA Mini Kit (Thermo Fisher Scientific K182002, Waltham, MA, USA) following the manufacturer’s instructions with some modifications.

### 4.5. Real-Time PCR

The real-time polymerase chain reaction was carried out in a LightCycler^®^ 96 (Roche Life Science) and performed as previously described [40] with some modifications. Real-time PCR for simultaneous detection and quantification of *L. infantum* kinetoplast minicircle DNA was performed using the primers LEISH-1 (5′-GGCGTTCTGCGAAAACCG-3′), LEISH-2 (5′-AAAATGGCATTTTCGGGCC-3′), and TaqMan probe 5′FAM-TGGGTGCAGAAATCCCGTTCA-3′- BHQ1. PCR amplification was carried out in 20 μL reactions containing (final concentration) SsoAdvanced Universal Probes Supermix (Biorad, Hercules, CA, USA), 0.3 μM of each primer, 0.25 μM QLeish Probe and 2 μL of extracted DNA at 10 ng/μL. The conditions were set as follows: UNG step at 50 °C for 150 sec, initial denaturation for 10 min at 95 °C, 40 cycles of denaturation at 95 °C for 15 sec and annealing-polymerization at 60 °C for 35 sec. A standard curve was constructed using 10-fold serially diluted *L. infantum* parasite DNA corresponding to 1 × 10^6^ to 1 parasite per mL. A threshold cycle value (Ct) was calculated for each sample by determining the point at which the fluorescence exceeded the threshold limit. A standard curve was obtained by plotting the Ct values against each standard of known concentration parasite DNA. (Appendix A). The samples that were showed a parasite load of ≥1 parasites per mL were considered positive.

### 4.6. Cloning and Plasmid Generation

The qPCR was conducted on total DNA extracted from the cultivated *L. infantum* and the target for the amplification was the 120-bp fragment in the constant region of the kinetoplast minicircle. The 3′ A-tailed fragments, were generated by AmplyTaq Gold, polymerase (Thermo, Waltham, MA, USA) using the primers: 120 iner forw. (TCCCGTTCATTTTTGGCCCG) and 120 Rev (ACCCCCAGTTTCCCGCC). The conditions were settled as follows: denaturation at 95 °C for 1 min, annealing at 53 °C for 1 min, and extension at 72 °C for 1 min. A 7 min extension period at 72 °C was added after 40 cycles. After electrophoresis of PCR products on 2% agarose gel, the amplified products were purified using “Illustra GFX PCR DNA and gel band Purification kit” (GE Healthcare, Chicago, IL, USA), and specthrotometrically quantified. Various recombinant plasmid quantities (10, 20, and 40 ng) were mixed and incubated with 50 ng pGem-T Easy Vector Systems (Promega, Madison, WI, USA). The reactions were settled up in 10 μL total volume, in particular 5μL Rapid Ligation Buffer, 1 μL pGem Vector, 1 μL T4 DNA Ligase and nuclease-free water. At the same time, a positive control, using Control DNA insert, and a background Control, without DNA, were employed. The resulting construct was transformed into competent cell JM109 High-Efficiency Competent Cells (Promega, Madison, WI, USA). 50μL of cells were transferred into a sterile tube containing 2 μL of the ligation reaction. It gently flicked and placed on ice for 20′. Successively, cells were heat-shocked for 60 s in a water bath at exactly 42 °C and immediately were returned to the ice for 2′. At the tube containing transformed cells, 950 μL of room-temperature SOC medium were added and the tube was placed for two hours at 37 °C with shaking (150 rpm). Different amounts of SOC medium containing transformed cells (250 μL, 100 μL, 50 μL, and 25 μL) were collected and cultured in a Petri dish containing LB/ampicillin/IPTG/X-Gal medium. The plate was incubating overnight at 37 °C. A number of white colonies were selected and grown on LB broth. After 1 night the bacteria were centrifuged to pellet and recovery the recombinant plasmids using a plasmid miniPrep kit (Qiagen, Hilden, Germany) for the extraction. The presence of the fragment into the vector was checked by PCR with specific primers and the amplicon was checked by size controlling on agarose gel and sequencing on 3130 Thermo scientific genetic analyzer. Forward and reverse sequencing reactions were conducted by using the “big dye sequencing kit” (Thermo, Waltham, MA, USA); the products were purified by the X-terminator kit (Thermo, Waltham, MA, USA). The *L. infantum* kDNA sequence was registered on a gene bank with the code: AJ131633.1. [74].

A real-time test was employed both on the recombinant plasmid and on the DNA extracted from the cultivated strain to compare the signals of the respective serial dilutions. As the first step, plasmids containing the insert were quantified and 7 different 1:10 dilutions were performed to construct the standard curves for the qPCR, starting from 100 ng/μL. The concentration of the plasmid was measured using a Qbit spectrophotometer (Thermofisher) and the corresponding copy number was calculated using the following equation [78]:
Plasmid copy = [6.02 × 10^23^ (copy/mol) × DNA amount (g)]/[DNA length (dp) × 660 (g/mol/dp)].



Statistical analysis was based on the Ct values obtained for 3 times repetitions at each point.

### 4.7. Clinical Sensitivity, Specificity & Statistical Analysis

Specificity was verified testing a collection panel of DNAs belonging to different *Leishmania* species: *L. aethiopica* (MHOM/ES/72/L100), *L. braziliensis* (MHOM/BR/75/M2904), *L. donovani* (MHOM/IN/80/DD8), *L. major* (MHOM/SU/73/5ASKH), *L. tropica* (MHOM/SU/74/K27) and *L. amazonensis* (IFLA/Br/67/PH8). Assay specificity was also tested in the presence of DNA from pathogens other than *Leishmania* such as *Trypanosoma cruzi*, *Ehrlichia*, *Babesia*, and *Theileria*. Sensitivity was determined using 7-fold serial dilutions of the standard from 10^6^ to 10^−1^ parasites for mL. The clinical sensitivity and specificity of the *Leishmania* TaqMan qPCR assay for detecting *Leishmania* parasites were calculated considering clinical confirmation with the conventional diagnosis “*gold standard*” for VL, CL, CanL, and FeL. Skin lesion tissues, bone marrow, blood samples, and popliteal lymph nodes were analyzed using the designed qPCR assay for evaluating its sensitivity and specificity for leishmaniasis.

## Figures and Tables

**Figure 1 pathogens-10-00865-f001:**
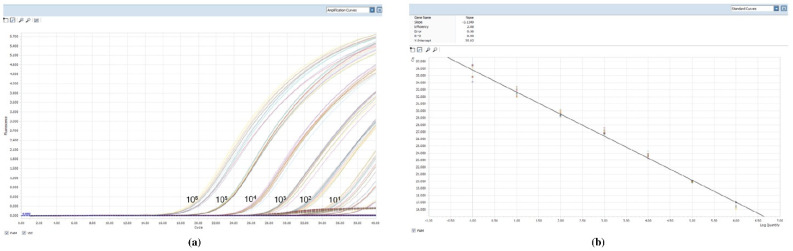
Technical performance and range of detection of the *Leishmania infantum* q PCR assay: in (**a**) DNA was extracted from serial dilutions of cultured *L. infantum*, ranging from 1 × 10^6^ to 1 *Leishmania*/mL. Amplification curves are shown for each dilution, with each parasite concentration; (**b**) the mean Ct values are plotted from ten tested against serial dilutions containing *L. infantum* per reaction. Each point represents the Ct of an individual sample, with the plot of Ct values and parasite equivalent fitting a linear function (R2 _0.99).

**Figure 2 pathogens-10-00865-f002:**
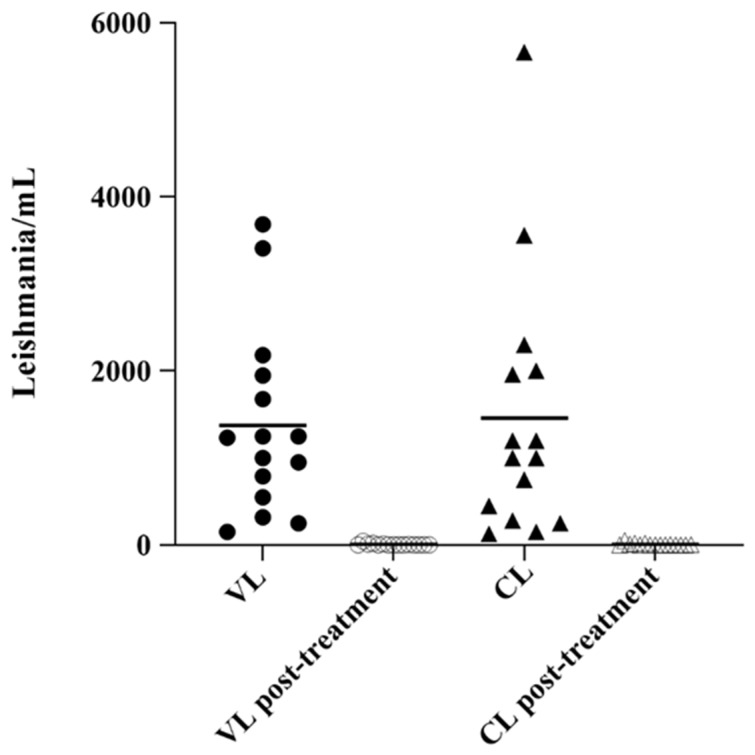
Scatter plot showing parasite load in the bone marrow and tissue/skin biopsy in VL and CL samples. Parasite load in VL bone marrow (parasites/mL blood) and in lesion tissues CL samples (parasites/million nucleated cells) were determined by real-time PCR, in pre and post-treatment. The horizontal line represents the mean parasite load.

**Table 1 pathogens-10-00865-t001:** Results from serological, parasitological, microscopy-histology, and molecular assays, distributed according to clinical status.

Clinical Presentation		Conventional Diagnosis (IFAT, PC, MH)			Real-Time PCR (qPCR) kDNA
	Clinical Samples	IFAT Positive/Total (%)	PC Positive/Total (%)	MH Positive/Total (%)	Positive/Total (%)
VL	Peripheral blood/serum	13/15 (86.6)	4/15 (26.6)	10/15 (66.6)	13/15 (86.6)
	Bone marrow	n.a.	8/15 (53.3)	13/15 (86.6)	15/15 (100)
CL	Tissue/skin biopsy	n.a.	9/15 (60)	10/15 (66.6)	15/15 (100)
CanL	Peripheral blood/serum	19/30 (63.3)	5/30 (16.6)	5/30 (16.6)	19/30 (63.3)
	Popliteal lymph node	n.a.	16/30 (53.3)	17/30 (56.6)	30/30 (100)
FeL	Peripheral blood/serum	8/20 (40)	0/20 (0)	0/20 (0)	1/20 (5)
	Popliteal lymph node	n.a.	1/20 (5)	1/20 (5)	20/20 (100)

IFAT, indirect immunofluorescent assay; PC, Parasite culture; MH, microscopy-histology; qPCR, quantitative PCR; VL, human visceral leishmaniasis; CL, human cutaneous leishmaniasis; CanL, canine leishmaniasis; FeL, feline leishmaniasis; n.a., not applicable.

**Table 2 pathogens-10-00865-t002:** Species, clinical status, and results of conventional and molecular methods in symptomatic and asymptomatic dogs and cats.

			Conventional Methods			Molecular Method	
Species	Clinical Status	Sample ID	IFAT Titer	Parasite Culture	Microscopy-Histology	qPCR Blood (*Leishmania*/mL)	qPCR Lymph Node (*Leishmania*/mL)
Canine	Symptomatic	1	1:160	Blood − Lymph node −	Blood − Lymph node −	45	540
		2	1:5120	Blood + Lymph node +	Blood + Lymph node +	590	15,680
		3	1:640	Blood − Lymph node +	Blood − Lymph node +	30	970
		4	1:1280	Blood − Lymph node +	Blood − Lymph node +	120	7838
		5	1:320	Blood − Lymph node +	Blood − Lymph node +	75	660
		6	1:2560	Blood − Lymph node +	Blood − Lymph node +	60	9275
		7	Negative	Blood − Lymph node −	Blood − Lymph node −	Negative	50
		8	1:80	Blood − Lymph node −	Blood − Lymph node −	Negative	100
		9	1:5120	Blood + lymph node +	Blood + Lymph node +	645	24,890
		10	1:160	Blood − Lymph node +	Blood − Lymph node +	10	390
		11	1:160	Blood − Lymph node +	Blood − Lymph node +	20	460
		12	1:640	Blood − Lymph node +	Blood − Lymph node +	285	5300
		13	1:320	Blood − Lymph node +	Blood − Lymph node +	230	910
		14	1:2560	Blood − lymph node +	Blood − Lymph node +	185	19,020
		15	1:2560	Blood + lymph node +	Blood+ Lymph node +	580	17,535
		16	1:320	Blood − Lymph node −	Blood − Lymph node +	150	4230
		17	1:1280	Blood − Lymph node +	Blood − Lymph node +	320	12,545
		18	1:160	Blood − Lymph node −	Blood − Lymph node −	35	850
		19	1:5120	Blood + lymph node +	Blood + Lymph node +	495	1,000,000
		20	1:1280	Blood + Lymph node +	Blood + Lymph node +	320	6710
	Asymptomatic	21	Negative	Blood − Lymph node −	Blood − Lymph node −	Negative	10
		22	1:40	Blood – Lymph node −	Blood − Lymph node −	Negative	20
		23	1:160	Blood − Lymph node +	Blood − Lymph node −	10	120
		24	1:40	Blood − Lymph node −	Blood − Lymph node −	Negative	15
		25	1:80	Blood − Lymph node −	Blood − Lymph node −	Negative	20
		26	Negative	Blood − Lymph node −	Blood − Lymph node −	Negative	15
		27	1:40	Blood − Lymph node −	Blood − Lymph node −	Negative	30
		28	1:40	Blood − Lymph node −	Blood − Lymph node −	Negative	65
		29	1:80	Blood − Lymph node −	Blood − Lymph node −	Negative	80
		30	1:80	Blood − Lymph node −	Blood − Lymph node −	Negative	35
Feline	Symptomatic	31	1:80	Blood − Lymph node −	Blood − Lymph node −	Negative	5
		32	1:80	Blood − Lymph node −	Blood − Lymph node −	Negative	10
		33	1:160	Blood − Lymph node −	Blood − Lymph node −	Negative	20
		34	1:160	Blood − Lymph node −	Blood − Lymph node −	Negative	5
		35	Negative	Blood − Lymph node −	Blood − Lymph node −	Negative	10
		36	1:640	Blood − Lymph node +	Blood − Lymph node +	250	980
		37	Negative	Blood − Lymph node −	Blood − Lymph node −	Negative	20
		38	Negative	Blood − Lymph node −	Blood − Lymph node −	Negative	25
		39	1:80	Blood − Lymph node −	Blood − Lymph node −	Negative	30
		40	1:160	Blood − Lymph node −	Blood − Lymph node −	Negative	55
		41	1:320	Blood − Lymph node −	Blood − Lymph node +	10	150
		42	1:80	Blood − Lymph node −	Blood − Lymph node −	Negative	15
		43	1:80	Blood − Lymph node −	Blood − Lymph node −	Negative	10
		44	1:160	Blood − Lymph node −	Blood − Lymph node −	Negative	35
		45	1:80	Blood − Lymph node −	Blood − Lymph node −	Negative	10
		46	Negative	Blood − Lymph node −	Blood − Lymph node −	Negative	5
		47	1:160	Blood − Lymph node −	Blood − Lymph node −	Negative	50
		48	Negative	Blood − Lymph node −	Blood − Lymph node −	Negative	5
		49	1:320	Blood − Lymph node −	Blood − Lymph node −	5	350
		50	Negative	Blood − Lymph node −	Blood − Lymph node −	Negative	60

IFAT, indirect immunofluorescent assay; qPCR, quantitative PCR.

## Data Availability

The study did not report any data.

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
