# Peer review of "Molecular Diagnosis of Leishmaniasis: Quantification of Parasite Load by a Real-Time PCR Assay with High Sensitivity"

_pathogens, 2021, doi:10.3390/pathogens10070865_

Round 1

Reviewer 1 Report

Unfortunately, the manuscript “Molecular diagnosis of leishmaniasis: quantification of parasite load by a Real-Time PCR assay with high sensitivity” could not be accepted for publication because doesn’t give any interesting, new, or useful information about the use of qPCR in leishmaniasis. Furthermore, the introduction needs to shorter and focused only in molecular diagnosis of leishmaniasis, and several references published previously about the use of qPCR in leishmaniasis are missing.

Author Response

Thank you for giving me the opportunity to submit a revised draft of my manuscript titled " Molecular diagnosis of leishmaniasis: quantification of parasite load by a Real-Time PCR assay with high sensitivity.”. We appreciate the time and effort that you have dedicated to providing your valuable feedback on manuscript. We have been able to incorporate changes to reflect most of your suggestions.

  • According to the observation of VL and CL patients, we included the treatment, dosage and results obtained:

In the section “results” we included “During weekly follow-up of VL and CL patients, molecular investigations performed on bone marrow and tissue/ skin biopsy showed a progressive reduction of parasite load (data not shown). Figure 2 showed the parasite load by real-time PCR in human clinical samples before starting treatment and one day after the last dose of treatment.” (Lines 172-176)

In the section “Material and methods” we included: “Patients of VL were given treatment with liposomal amphotericin B (AmBisome®) 3 mg/kg administered intravenous route (IV) once daily days 1–5, 14 and 21. CL cases were treated with pentavalent antimonial (SbV) therapy, the standard daily dose was 20 mg of SbV per kg, administered IV or intramuscular route (IM). The diagnosis of VL and CL was made by microscopic identification of the intracellular form (amastigote) in stained sections from bone marrow/skins biopsy, real-time PCR and serological tests. For some patients, adjustment of the daily dose or the duration of therapy could be indicated. No standard treatment regimen had been established, various regimens had been used depending in part on the size, characteristics of the lesions and immunological status. All patients were then monitored for follow-up visits at weekly intervals, at which clinical manifestation were reassessed and biological samples were collected and evaluated for conventional and molecular diagnosis. Clinical samples were taken before starting treatment and one day after the last dose of treatment for qPCR evaluation.” (Lines 419-433)

  • According the observation of the aim of discriminating between Leishmania infantum and the other species:

In the section “discussion” we included “However, future studies should be carried out to further explore this issue. There is no single molecular gold standard diagnostic test, for all regions of the world because there are changes in the DNA sequences within a species according to their origin. The qPCR assays for leishmaniasis that amplify species-specific DNA sequences are not appropriate for regions with sympatric Leishmania species. In fact, even in areas where Leishmania species are known, the presence of poly-morphisms and variations in copy number of target sequences (e.g. minicircle classes), could interfere with the performance of qPCR assays [21]. To rule out false negatives or co-infections with other Leishmania species, the species-specific qPCR should be combined with a genus-specific assay (broad range assay). Therefore, approaches based on initial amplification of genus-specific sequences followed by assays for species differentiation/typing should be useful [21].” (Lines 271-283)

  • We changed the legend of Figure 1 removing “depicted by a differing colour”. (Line 138)
  • We changed the resolution of Figure 1
  • In Table 1 we included “n.a., not applicable.” and we included “VL, human visceral leishmaniasis; CL, human cutaneous leishmaniasis”. We removed “showed titers of ≥1:160 were considered positive”.
  • Table 2 does not include human leishmaniasis cases as it describes the diagnosis of CanL and FeL, comparing for each patient (asymptomatic and symptomatic) both different biological matrices and different methods (conventional and molecular assays). Human leishmaniasis and the concept of therapeutic monitoring has been addressed in a chapter of the manuscript and in figure 2. if it is deemed necessary, we could integrate human patients also in table 2.
  • We removed table 3
  • We removed “Table 3 showed the VL, CL, CanL and FeL patients included in the study.” In “materials and methods” section
  • In the supplementary tables we changed The column O in “coefficient of variation of ct values (%)”.
  • We changed Leishmania in italics throughout the text.
  • We revised the bibliography.

In the section “introduction” we removed “The natural cycle involves phlebotomine sandfly vectors transmitting the parasite to the vertebrate host. The geographic distribution and the specie-specific vectorial capacity of the phlebotomine influence the epidemiology of the disease [2].” and “Amastigote and promastigote retained hundreds of DNA maxicircles and thousands of DNA minicircles respectively, both with a conserved origin of replication encoding guide RNA sequence for RNA editing [37–39]. Because the kDNA target copy number, the qPCR became a very sensitive test useful for Leishmania detection and absolute quantification [21].”

We included: “Other assays were developed to estimate the parasite load through absolute or relative quantification.” and “Various chromosomal and kinetoplast DNA sequences has been valuated as target for diagnostic systems and the extra-chromosomal circles showed always the best sensitivity and represent currently the most employed target with worldwide diffusion [39]. The use of high-copy 18S and minicircle kDNA sequences as a target usually allows amplification at the genus or subgenus level, due to their conserved sequences or to the heterogeneity of minicircle classes [21].”  adding a more recent bibliography (39.    Ceccarelli, M.; Buffi, G.; Diotallevi, A.; Andreoni, F.; Bencardino, D.; Vitale, F.; Castelli, G.; Bruno, F.; Magnani, M.; Galluzzi, L. Evaluation of a KDNA-Based QPCR Assay for the Detection and Quantification of Old World Leishmania Species. Microorganisms 2020, 8, doi:10.3390/microorganisms8122006.” (Lines 71-89)

We look forward to hearing from you in due time regarding our submission and to respond to any

further questions and comments you may have.

Sincerely,

Federica Bruno

Reviewer 2 Report

The work reported in this manuscript represents an interesting contribution to the field of molecular detection of leishmaniasis, although other studies have reported a detection limit of Leishmania spp. of 0.1 paramites/ml. (Mary, C. et al., 2006. Am. J. Trop. Med. Hyg. 75 (5), 858–863.), (M.C. Guillén et al., Heliyon 6 (2020) e03940).

The manuscript is well structured as it describes the process of fine-tuning the technique followed by its application. Arguably, the most valuable contribution of the technique would lie in its capacity to evaluate the treatment and determine the eradication of leishmaniasis. In this regard, more details are required about how the treatment is carried out and its follow up. Data in Figure 2 indicate that 15 cases of VL and 15 of CL were studied, but there is no other information about these cases other than the limited data in Table 1.

To evaluate the usefulness of the technique, the patients need to be followed up. Therefore, information is needed about the treatment each one received, the dosage regime, the molecular, serological, and microscopic monitoring, the time elapsed until the negative PCR, and if the result is supported by other techniques.

In addition, the aim of discriminating between Leishmania infantum and the other species seems not to have been met. It would appear that a positive result confirms L. infantum but a negative result combined with positive serologies indicates that another Leishmania species cannot be ruled out, as they are not excluded by this technique. Is this the case? I think that to be able to assess the real contribution of the technique, this aspect should be described in more detail. In this globalized world, diseases endemic to specific areas can have an impact elsewhere, and it cannot be assumed that the non-detection of L. infantum DNA implies an absence of leishmaniasis, especially in humans.

The legend of Figure 1 states: Amplification curves are shown for each dilutions, with each parasite concentration depicted by a differing color. However, instead, the different colours seem to indicate the different ranges of dilutions tested. The legend should be corrected accordingly. Additionally, the legends of both images (Figure 1a and 1b) are too small to read, as the resolution is lost when amplified.

In the tables it should be clarified that VL and CL refer to human leishmaniasis.

In Table 1, please explain what “n.a.” stands for. The footnote includes the IFAT cut-off, but this information belongs in the methodology section, not the table. The IFAT cut-off should also be given for the qPCR.

Why does Table 2 not include the results of the 15 cases of human leishmaniasis included in Table 1?

Table 3 could be deleted, as it does not provide any additional information to the text and repeats data of Table 2.

In the supplementary tables, column O (coefficient of variation of the parasite number (%)), specifically the rows between 3 and 10, should be entitled: coefficient of variation of Ct values (%).

Throughout the text, “Leishmania” should be in italics.

The bibliography needs a thorough revision, as many references are not well cited. The main error is that that the authors are cited only by the initials of their names and surnames.

Author Response

Thank you for giving me the opportunity to submit a revised draft of my manuscript titled " Molecular diagnosis of leishmaniasis: quantification of parasite load by a Real-Time PCR assay with high sensitivity.”. We appreciate the time and effort that you have dedicated to providing your valuable feedback on manuscript. We have been able to incorporate changes to reflect most of your suggestions.

  • According to the observation of VL and CL patients, we included the treatment, dosage and results obtained:

In the section “results” we included “During weekly follow-up of VL and CL patients, molecular investigations performed on bone marrow and tissue/ skin biopsy showed a progressive reduction of parasite load (data not shown). Figure 2 showed the parasite load by real-time PCR in human clinical samples before starting treatment and one day after the last dose of treatment.” (Lines 172-176)

In the section “Material and methods” we included: “Patients of VL were given treatment with liposomal amphotericin B (AmBisome®) 3 mg/kg administered intravenous route (IV) once daily days 1–5, 14 and 21. CL cases were treated with pentavalent antimonial (SbV) therapy, the standard daily dose was 20 mg of SbV per kg, administered IV or intramuscular route (IM). The diagnosis of VL and CL was made by microscopic identification of the intracellular form (amastigote) in stained sections from bone marrow/skins biopsy, real-time PCR and serological tests. For some patients, adjustment of the daily dose or the duration of therapy could be indicated. No standard treatment regimen had been established, various regimens had been used depending in part on the size, characteristics of the lesions and immunological status. All patients were then monitored for follow-up visits at weekly intervals, at which clinical manifestation were reassessed and biological samples were collected and evaluated for conventional and molecular diagnosis. Clinical samples were taken before starting treatment and one day after the last dose of treatment for qPCR evaluation.” (Lines 419-433)

  • According the observation of the aim of discriminating between Leishmania infantum and the other species:

In the section “discussion” we included “However, future studies should be carried out to further explore this issue. There is no single molecular gold standard diagnostic test, for all regions of the world because there are changes in the DNA sequences within a species according to their origin. The qPCR assays for leishmaniasis that amplify species-specific DNA sequences are not appropriate for regions with sympatric Leishmania species. In fact, even in areas where Leishmania species are known, the presence of poly-morphisms and variations in copy number of target sequences (e.g. minicircle classes), could interfere with the performance of qPCR assays [21]. To rule out false negatives or co-infections with other Leishmania species, the species-specific qPCR should be combined with a genus-specific assay (broad range assay). Therefore, approaches based on initial amplification of genus-specific sequences followed by assays for species differentiation/typing should be useful [21].” (Lines 271-283)

  • We changed the legend of Figure 1 removing “depicted by a differing colour”. (Line 138)
  • We changed the resolution of Figure 1
  • In Table 1 we included “n.a., not applicable.” and we included “VL, human visceral leishmaniasis; CL, human cutaneous leishmaniasis”. We removed “showed titers of ≥1:160 were considered positive”.
  • Table 2 does not include human leishmaniasis cases as it describes the diagnosis of CanL and FeL, comparing for each patient (asymptomatic and symptomatic) both different biological matrices and different methods (conventional and molecular assays). Human leishmaniasis and the concept of therapeutic monitoring has been addressed in a chapter of the manuscript and in figure 2. if it is deemed necessary, we could integrate human patients also in table 2.
  • We removed table 3
  • We removed “Table 3 showed the VL, CL, CanL and FeL patients included in the study.” In “materials and methods” section
  • In the supplementary tables we changed The column O in “coefficient of variation of ct values (%)”.
  • We changed Leishmania in italics throughout the text.
  • We revised the bibliography.

In the section “introduction” we removed “The natural cycle involves phlebotomine sandfly vectors transmitting the parasite to the vertebrate host. The geographic distribution and the specie-specific vectorial capacity of the phlebotomine influence the epidemiology of the disease [2].” and “Amastigote and promastigote retained hundreds of DNA maxicircles and thousands of DNA minicircles respectively, both with a conserved origin of replication encoding guide RNA sequence for RNA editing [37–39]. Because the kDNA target copy number, the qPCR became a very sensitive test useful for Leishmania detection and absolute quantification [21].”

We included: “Other assays were developed to estimate the parasite load through absolute or relative quantification.” and “Various chromosomal and kinetoplast DNA sequences has been valuated as target for diagnostic systems and the extra-chromosomal circles showed always the best sensitivity and represent currently the most employed target with worldwide diffusion [39]. The use of high-copy 18S and minicircle kDNA sequences as a target usually allows amplification at the genus or subgenus level, due to their conserved sequences or to the heterogeneity of minicircle classes [21].”  adding a more recent bibliography (39.    Ceccarelli, M.; Buffi, G.; Diotallevi, A.; Andreoni, F.; Bencardino, D.; Vitale, F.; Castelli, G.; Bruno, F.; Magnani, M.; Galluzzi, L. Evaluation of a KDNA-Based QPCR Assay for the Detection and Quantification of Old World Leishmania Species. Microorganisms 2020, 8, doi:10.3390/microorganisms8122006.” (Lines 71-89)

We look forward to hearing from you in due time regarding our submission and to respond to any

further questions and comments you may have.

Sincerely,

Round 2

Reviewer 1 Report

I suggest two minor changes: a) to add a publication that gives the same clinical information that it’s not in the references (Martínez et al. Parasites & Vectors 2011, Canine leishmaniasis: the key points for qPCR result interpretation), and b) in the new text added by the authors the word “Leishmania” needs to be in cursive.

Author Response

Dear,

We have proceeded to insert the suggested reference and  we have changed "Leishmania" in italics.

regards